# Relationships among Physical Activity, Physical Function, and Food Intake in Older Japanese Adults Living in Urban Areas: A Cross-Sectional Study

**DOI:** 10.3390/geriatrics8020041

**Published:** 2023-04-03

**Authors:** Takashi Fushimi, Kyoko Fujihira, Hideto Takase, Masashi Miyashita

**Affiliations:** 1Biological Science Research, Kao Corporation, 2-1-3 Bunka, Sumida-ku, Tokyo 131-8501, Japan; 2Graduate School of Sport Sciences, Waseda University, 2-579-15 Mikajima, Tokorozawa 359-1192, Saitama, Japan; 3Faculty of Sport Sciences, Waseda University, 2-579-15 Mikajima, Tokorozawa 359-1192, Saitama, Japan; 4School of Sport, Exercise and Health Sciences, Loughborough University, Loughborough LE11 3TU, Leicestershire, UK; 5Department of Sports Science and Physical Education, The Chinese University of Hong Kong, Shatin, New Territories, Hong Kong

**Keywords:** physical activity, physical function, food, nutrition, older Japanese adults

## Abstract

Japan is experiencing a super-ageing society faster than anywhere else in the world. Consequently, extending healthy life expectancy is an urgent social issue. To realize a diet that can support the extension of healthy life expectancy, we studied the quantitative relationships among physical activities (number of steps and activity calculated using an accelerometer), physical functions (muscle strength, movement function, agility, static balance, dynamic balance, and walking function), and dietary intake among 469 older adults living in the Tokyo metropolitan area (65–75 years old; 303 women and 166 men) from 23 February 2017 to 31 March 2018. Physical activities and functions were instrumentally measured, and the dietary survey adopted the photographic record method. There was a significant positive association (*p* < 0.05) between physical activities (steps, medium-intensity activity, and high-intensity activity) and physical functions (movement function, static balance, and walking function), but no association with muscle strength. These three physical functions were significantly positively correlated with intake of vegetables, seeds, fruits, and milk; with magnesium, potassium, and vitamin B6; and with the dietary fibre/carbohydrate composition ratio (*p* < 0.05). Future intervention trials must verify if balancing diet and nutrition can improve physical activities in older adults through increased physical functions.

## 1. Introduction

Japan is experiencing a super-ageing society at a faster rate than anywhere else in the world; approximately 28.4% of the total population is aged 65 years or older, and more than 18 million people are aged 75 years or older [1]. In such a social situation, there is a need to respond to the decrease in the working-age population (15–65 years old) and the increased burden of nursing care. In addition, from a medical perspective, reducing the burden of nursing care by extending healthy life expectancy is an urgent issue. Thus, a national health promotion programme—Health Japan 21 (the second term)—was formulated [2], given that metabolic syndrome and locomotive syndrome were observed as a response to cardiovascular diseases and decreased physical function due to ageing.

Concerning abdominal fat, the accumulation of visceral fat is a higher risk factor for metabolic syndrome than subcutaneous fat [3]. Thus, we focused on visceral fat accumulation [4], which is key to the diagnostic criteria for metabolic syndrome, and examined the relationship between diet and lifestyle. In a previous study [5], we confirmed that dietary quality, quantity, time, night meals, fast eating, and inactivity are related to visceral fat accumulation. Among these factors, improving dietary quality might suppress visceral fat accumulation. Further examination revealed that the dietary composition of protein (kcal)/lipid (kcal) = 1.0, dietary fibre (g)/carbohydrate (g) > 0.063, and ω3 (mg)/lipid (g) > 0.054 might reduce the accumulation of visceral fat. In another clinical intervention study, this dietary composition was shown to reduce visceral fat accumulation (the SMART WASYOKU^®^ cuisine, Kao Corporation, Tokyo, Japan) [6].

Physical activity and exercise are involved in extending healthy life expectancy [7,8], especially among those with a high physical activity level [9]. Intensity and duration of physical activity are associated with balance, flexibility, and walking function in older people [10], and the relationships between skeletal muscle strength, agility, and physical activity have also been reported [11,12]. Thus, the need for physical activity guidelines to enhance physical functions to support activities of daily living in older adults has been advocated [13].

The relationships among physical function, frailty, degree of care required, and lifestyle in older Japanese adults have been reported in several regions of Japan [14,15,16]. Cohort studies have reported the relationship between physical activity and dietary content based on qualitative measurement [17,18,19], where either diet or physical activity or function was evaluated using a qualitative assessment method. Many dietary surveys adopted a food frequency questionnaire, the weakness of which is a low quantitative accuracy [20]. Consequently, no research studies have performed quantitative assessments of dietary content, physical activity, and function. Therefore, we conducted a cross-sectional study among older Japanese adults to quantitatively evaluate the relationships among physical activities, physical functions, and dietary content in order to propose dietary habits that could enhance physical functions to support the activities of older adults. In this study, instrumental measurements were used for physical activities and functions, and the dietary survey, in particular, adopted the photographic record method for quantitative understanding.

## 2. Materials and Methods

### 2.1. Participants

This study was conducted between 23 February 2017 and 31 March 2018. Older Japanese adults aged 65–75 years and living in the suburban areas of west Tokyo (Nishitokyo City, Koganei City, and Kokubunji City in Tokyo, and Tokorozawa City, Iruma City, and Sayama City in Saitama Prefecture) were recruited through Silver Human Resource Centres or local communities. The inclusion criteria were the ability to visit the recruitment and measurement venue on their own; the ability to take photos of all meals (including noshes and late-night snacks) for three days; the ability to wear an accelerometer on their waist for a week; and the ability to follow the researchers’ instructions. The exclusion criterion was unsuitability for participation as judged by the principal investigator, that is, it would be difficult to measure their physical functions.

Participants were 473 men and women who voluntarily expressed a willingness to participate. Based on a multiple linear regression analysis of the 36 survey items (characteristics, physical activities, physical functions, and food intake), we aimed to include at least 360 participants, or 10 times the number of survey items [21]. We explained the examination to the participants at a briefing session and obtained their written informed consent to participate in this research. After excluding 4 people who did not participate in the measurements, 469 people who completed the study were included in the analysis.

The study procedure was approved by two ethics committees (Waseda University’s Ethical Review Committee on Human Research and Kao Corporation’s Institute of Biological Sciences Research Ethics Review Committee) and implemented in accordance with the Declaration of Helsinki. This study was also registered in advance with the University Hospital Medical Information Centre (UMIN), a system for registering clinical trials (ID: UMIN000026007).

### 2.2. Schedule

At the briefing session, the study purpose, methodology, and protocol were explained to the participants, and they were informed of their right to withdraw from the study at any time. Written informed consent for participation was obtained from all individual participants. After agreeing to participate, height and weight were measured. Subsequently, the questionnaires, a disposable camera, and an accelerometer with ID registration were distributed individually. More than one week later, the measurement meeting was held on a morning. By the day of the measurement session, the participants had already completed the questionnaires, surveyed their diet, and wore their accelerometers. At the reception on the day of the measurement, the questionnaires, camera, and accelerometer were collected, and blood pressure was measured. Blood sampling and visceral fat measurement were performed by nurses. Afterwards, physical functions were measured by skilled staff who took great care to prevent the participants from falling (Figure 1).

### 2.3. Questionnaire Survey

After agreeing to participate in the examination at the briefing session, the following questionnaire survey was administered and collected on the day of the measurement meeting: inquiries about physical conditions and medical history; the International Physical Activity Questionnaire (Japanese version), which confirms physical activity; and dietary and lifestyle questionnaires.

### 2.4. Dietary Survey

Photo recording was adopted as the quantitative assessment method. The participants took photographs of all meals, including snacks, for three days, using a disposable film camera with lens and flash (Utsurundesu; FUJIFILM Corporation, Tokyo, Japan) [5]. In addition, a description of all meals was prepared. Meal content was analysed by two independent dietitians skilled in meal photo analysis while collating the meal descriptions. The analysis was performed using a dietary composition analytical software (Healthy Maker Pro 501; Mushroom Soft, Co., Ltd., Okayama, Japan) that contains a database of typically consumed Japanese meals and foods. Based on the results, the estimated intake of nutrients and foods was calculated based on the Standard Tables of Food Composition in Japan, using the residual method to adjust for total energy intake [22].

### 2.5. Anthropometric Measurements

Body mass was measured to the nearest 0.1 kg using a digital scale (Inner Scan 50; Tanita Corporation, Tokyo, Japan). Height was measured to the nearest 0.1 cm using a stadiometer (YS-OA; AS ONE Corporation, Osaka, Japan). Body mass index (BMI) was calculated as weight in kilogrammes divided by the square of height in metres. Arterial blood pressure was measured from the left arm in a sitting position using a standard mercury sphygmomanometer (605P; Yagami Co., Ltd., Yokohama, Japan). Two measurements were taken, and the mean of these values was recorded. Visceral fat was measured using abdominal bioelectrical impedance analysis (EW-FA90; Panasonic Corporation, Osaka, Japan), which estimates the umbilical level of the visceral fat area. Waist circumference was measured in the standing position at the level of the umbilicus to the nearest 0.1 cm using a constant tension tape.

### 2.6. Physical Activity

The amount of physical activity and the number of walking steps were measured while the participants performed their usual activities, using an accelerometer (Lifecorder EX; Suzuken Co., Ltd., Nagoya, Japan) placed on the participants’ waist at all times while awake, except during swimming or bathing. The criteria for analysis were as follows: wearing of the accelerometer on their waist for a total duration of ≥4 days, including weekdays and holidays, for ≥10 h/day [23,24,25]. Exercise intensity was defined as follows: micro-exercise: <1.8, low intensity: 1.8–3.0, medium intensity: 3.0–6.0, and high intensity: ≥6.0 (unit: metabolic equivalents) [26]. The number of steps was calculated using a dedicated software (Lifelyzer 05 Coach; Suzuken Co., Ltd., Nagoya, Japan).

### 2.7. Physical Functions

#### 2.7.1. Muscle Strength

For grip strength, the average value measured on each arm was determined with a hand-held dynamometer (T.K.K.5401 Grip D; Takei Scientific Instruments Co., Ltd., Niigata, Japan). The grip bar was adjusted so that the second joints of the fingers were bent to grip the handle of the dynamometer. The participants stood upright, with the arm vertical and the dynamometer close to the body. The participants were then asked to squeeze the handgrip dynamometer as hard as possible. To determine knee extension muscle strength, the maximum voluntary contraction of the quadriceps muscle was measured for the dominant leg with a dynamometer force sensor (Locomo Scan; ALCARE Co., Ltd., Tokyo, Japan). First, the participants practised extending the knee isometrically by fixing the ankle with a belt of the device, and the assessor instructed the participants not to bend backwards and raise their hips. Second, the assessor measured the participants’ maximum quadriceps muscle strength for 10 s. The test was repeated twice and recorded as the average. Afterwards, the average absolute value, divided by body weight, was computed to a relative value.

#### 2.7.2. Movement Function

The degree of the locomotive syndrome was assessed using a locomotive functional scale questionnaire, the two-step test, and the standing movement test [27,28,29]. In the two-step test, the width that advanced two steps by the crotch as far as possible, while avoiding falls and staggering, was corrected by adopting height.

#### 2.7.3. Agility Function

The open–close stepping test was used. In the test, the inside and outside of the tape wire were opened and closed concurrently at intervals of 30 cm, and a step was taken. The number of times this could be stepped in 20 s was measured.

The participants sat in a chair with no armrests, with both feet placed in the centre of a simple measurement sheet (30 × 30 cm). The participants’ hands were used to hold both sides of the chair. As soon as the experimenter signalled the participants to start, the participants opened their legs and spread their feet as quickly as possible, touching the floor beside the sheet with the forefoot or the entire sole of the foot, and then quickly returned their feet and legs to the original position. This series of actions constituted one repetition; the experimenter counted how many repetitions the participants could perform in 20 s [30].

#### 2.7.4. Balance Function

The one-leg standing (OLS) test with eyes open was used to assess sedentary balance. The ability of the participants to stand on one leg with their eyes open was measured for up to 120 s in accordance with the procedures for physical tests for older adults designed by the Ministry of Education, Culture, Sports, Science and Technology [31].

The functional reach test was used to assess movement balance. This was performed by asking the participants to stand, raise one arm that was adjacent to the wall parallel to the floor (90 degrees anterior flexion), and then bend their body and arm as far forward as they were comfortable without losing their balance. The assessor recorded the distance from the starting position to as far forward the tip of the middle finger of one raised arm could reach in centimetres [32]. The test was repeated twice, and the average was recorded.

#### 2.7.5. Walking Function

The timed up-and-go (TUG) test was used to assess walking function. The participants were instructed to move from a seated position in an armless chair to a standing position, walk 3 m at their fastest safe pace, turn around, walk back to the chair, and sit down again. The assessor measured the time using a stopwatch from when the participants were asked to start the test to when they were seated again. The test was repeated twice, and the average was recorded [33,34].

### 2.8. Blood Analysis

The participants were asked to refrain from eating or drinking after 9:00 p.m. before the measurement day. Consequently, they fasted overnight for more than 12 h, and blood samples were collected. The levels of the following variables were measured: items related to glucose and lipid metabolism, and minerals.

### 2.9. Statistical Analysis

All statistical analyses were performed using SPSS (version 24.0 for Windows; SPSS, Inc., Armonk, NY, USA). The relationships between physical activities and physical functions were analysed using multiple regression. Foods, nutrients, and nutritional compositions related to physical function were extracted using a multiple regression analysis with a forward selection method. The relationships between serum magnesium (Mg) concentration and intake of Mg and fibre, and fibre intake/carbohydrate intake ratio (Fib/C), were similarly analysed. Individual statistical values were not calculated for items that could not be detected using the forward selection method, so they are indicated as ‘-’ in the tables. Since the number of female participants was nearly twice that of male participants, sex was adopted as a basic adjustment item using a forced entry method. Subsequently, sex, age, and BMI were used as adjustment items. To simply compare the results of the three adjustment patterns of the multiple regression analysis, *p*-value correction was not performed. Pearson’s product-moment correlation coefficient (r) was calculated to study the relationship between the Fib/C ratio and intake of vegetables, fruits, cereals, Mg, potassium (K), and vitamin B6 (VB6).

## 3. Results

The participants’ characteristics (*n* = 469) are shown in Table 1. The systolic blood pressure was more than 140 mmHg in both sexes, and the mean values of other characteristics were in the normal range, even though visceral fat accumulation was observed to be high in men (Table 1).

The number of steps per day, the grip strength, and the OLS test in men were higher than those in women, but there were no significant differences between the sexes (Table 2).

As shown in Table 3a, similar associations were found between medium-intensity exercise and the two-step test and between high-intensity exercise and the OLS test. In addition, micro-exercise intensity was significantly negatively related to the results of the two-step test. However, significant associations between physical activities and other physical functions (muscle strength, agility function, and movement balance) were not found. The results were similar regardless of the adjustment items (Table 3b,c).

Furthermore, the relationships between the results of the aforementioned physical function measurement tests (two-step, OLS, and TUG) and food intake were examined (Table 4a). Foods and nutrients that have a significant relationship with any of these physical functions are shown in Table 4. Vegetable intake was associated with an extension of standing time (OLS test; *p* = 0.023), while cereal and meat intakes were related to shorter standing times (*p* = 0.006 and *p* = 0.003, respectively). Vegetable and milk intakes were associated with faster movement (TUG test; *p* = 0.002 and *p* = 0.034, respectively); however, cereal intake was associated with a delay in movement (*p* = 0.026). Vegetable, seed, fruit, and beverage intakes were associated with longer strides (two-step test; *p* = 0.023, *p* = 0.015, *p* = 0.017, and *p* = 0.010, respectively), while egg intake was associated with shorter strides (*p* = 0.018). In terms of nutrients, magnesium (Mg) intake was positively related to the results of the OLS test, potassium (K) intake was positively related to the results of the TUG test, vitamin B6 (VB6) intake was positively related to the results of the two-step test, and dietary fibre intake/carbohydrate intake ratio (Fib/C) was positively related to the results of all three tests. The results were similar regardless of the adjustment items (Table 4b,c).

## 4. Discussion

This study examined the quantitative relationships among diet, physical functions, and physical activities in older Japanese adults. Significant positive associations between physical activities (i.e., steps, medium-intensity activity, and high-intensity activity) and physical functions (i.e., movement, static balance, and walking function) were found. The intake of certain foods, such as vegetables, fruits, and seeds, and their contained nutrients (Mg, K, VB6, and fibre) was positively correlated with these physical functions.

The average values of items other than systolic blood pressure were within the diagnostic criteria for metabolic syndrome (Table 1) [35]. Since blood pressure was measured immediately after arrival at the venue and at reception on the day of measurement, systolic blood pressure might be high. The average numbers of steps were higher than those of Japanese individuals of similar ages for both sexes (Table 2). There are reports suggesting that the average number of steps increases with the size of cities in Japan [36] and that the physical activities of urban dwellers are higher than those of rural dwellers [37]. Thus, it was concluded that the participants in this study lived active daily lives; however, their physical functions, such as muscle strength and balance, might not be superior to those of other individuals of the same age [38].

The physical functions that were significantly associated with the number of steps were sedentary balance (OLS test), walking function (TUG test), and movement function (two-step test). Medium- and high-intensity exercise was significantly related to movement function and sedentary balance, respectively, which is consistent with a previous report [10]. Thus, a significant association was found between physical activities (number of steps, medium-intensity exercise, and high-intensity exercise) and physical functions (sedentary balance, walking function, and movement function). In contrast, other physical functions, such as muscle strength and agility, were not associated with physical activity in this study. A high level of daily physical activity is reportedly related to high mobility function rather than muscle strength [39]. The current results are generally consistent with those of the aforementioned report. Steps are also a basic activity of daily life and an essential element in maintaining high physical activity for older adults [40]. Therefore, improvement of these three physical functions (sedentary balance, walking function, and movement function) may be important to support physical activity in older adults.

The intake of vegetables, fruits, and seeds was positively associated with these physical functions. In addition, the relationships between the intake of their contained nutrients (Mg, K, and VB6) and physical functions were similar (Table 4). These results are consistent with previous reviews based on qualitative studies using a food frequency questionnaire [19,41]. Mg is involved in muscle relaxation and contraction [42], and its supplementation reportedly improves physical function [43]. K is reportedly involved in neurotransmission and muscle contraction, and it has been recently reported that VB6 intake is associated with agility and mobility and that its contribution is derived from fruits [44]. In addition, Mg, K, and VB6 intakes are reported to be positively associated with physical activity [45]. Although the direct relationship is unclear, intakes of such foods and nutrients might play a role in changes in physical function.

In particular, besides vegetable intake, the Fib/C ratio, which is a component composition of nutrient intake, was positively correlated with all those three physical functions (sedentary balance, walking function, and movement function), as confirmed through the quantitative survey. Fibre intake is suggested to be associated with physical activity [46], but its relationship with physical functions is unknown. Fibre reportedly affects the absorption of various nutrients. The absorption of K and VB6 is primarily owing to passive transport; therefore, the effect of fibre intake is unknown. A report indicates that dietary fibre does not affect the bioavailability of VB6 in food [47]. Conversely, the enhancing effect of fibre on Mg absorption through active transportation has been reviewed at the level of animal experiments and clinical trials [48]. It was confirmed that the Fib/C ratio, rather than fibre and Mg intake, was significantly positively related with the blood Mg concentration based on the multiple regression analysis with a forward selection method (Table 5a–c). Thus, a high Fib/C ratio might indirectly affect physical function by increasing Mg absorption. In contrast, carbohydrate restriction has been reported to improve knee function in patients with type 2 diabetes [49], but the relationship between carbohydrate intake and the three physical functions is unclear. Fib/C is one of the dietary compositions of the SMART WASYOKU^®^ cuisine for the reduction of visceral fat accumulation [5,6]. This cuisine recommends the intake of vegetables, fruits, seaweed, beans, and fish while avoiding cereals, meats, and fats. In addition, the Fib/C ratio was positively correlated with the intake of vegetables, fruits, seeds, Mg, K, and VB6; however, it was negatively correlated with the intake of cereals (Pearson’s r = 0.536, 0.256, 0.293, 0.683, 0.736, 0.577, and −0.431, respectively; *p* < 0.001 for all). It was considered that some dietary habits that increased the Fib/C ratio were linked to the characteristics of the intake of such foods and nutrients. Considering these and the relationships between physical functions and foods in this study (Table 4), a high intake of vegetables and fruits and a low intake of cereals might be effective for a high Fib/C ratio to improve the three physical functions.

There was no significant relationship between muscle strength and the intake of three protein-rich foods (meat, milk, and egg), which was negatively related with sedentary balance, walking function, and movement function, respectively. This might be because the participants were suggested to live actively; however, the details were unclear.

Based on the current results, we elucidated the quantitative relationships among physical activities, physical functions, and food intake in older Japanese adults living in urban areas. Several limitations should be considered. First, this study was conducted following a cross-sectional design. Long-term observational studies are needed to examine diets that could improve physical functions and activities. Second, the participants were living in urban areas. Third, compared with the national average of steps in individuals of the same age, these participants were relatively active. Fourth, the sample size was small. Consequently, the relationships, as previously reported [11,12], between skeletal muscle strength, agility, and physical activities require further elucidation. Thus, it is necessary to include participants who live in rural areas and who participate in other activities with a large sample size.

In conclusion, we found significant relationships among the intake of certain foods and nutrients, physical functions, and physical activities in older Japanese adults. Future studies should conduct a clinical trial using foods (high intake of vegetables in particular, fruits, and seeds, and low intake of cereals) or nutrients (high intake of Mg, K, and VB6, and, especially, high Fib/C ratio) recognised from this study as possible intervention targets. Thus, we propose a diet that supports the movement and activities of the older adult population.

## Figures and Tables

**Figure 1 geriatrics-08-00041-f001:**
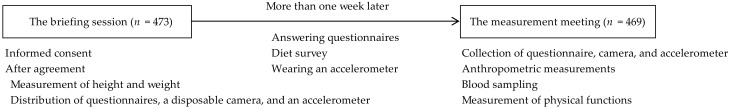
Study schedule.

**Table 1 geriatrics-08-00041-t001:** Physiological characteristics of the older Japanese adults living in urban areas.

	Total	Men	Women
Number	469	166	303
Age (years)	69.8 ± 2.9	70.2 ± 3.0	69.5 ± 2.9
Body mass index (kg/m^2^)	22.7 ± 3.8	23.5 ± 4.8	22.2 ± 2.9
Visceral fat area (cm^2^)	75.5 ± 38.2	94.8 ± 43.4	64.9 ± 30.2
Systolic blood pressure (mmHg)	141 ± 21	142 ± 22	140 ± 20
Diastolic blood pressure (mmHg)	81 ± 12	83 ± 12	80 ± 11
Fasting blood glucose (mmol/L)	5.58 ± 0.90	5.78 ± 0.91	5.48 ± 1.03
Triacylglycerol (mmol/L)	1.15 ± 0.75	1.27 ± 0.90	1.09 ± 0.64
LDL-cholesterol (mmol/L)	3.38 ± 0.79	3.16 ± 0.76	3.50 ± 0.88
HDL-cholesterol (mmol/L)	1.76 ± 0.43	1.61 ± 0.41	1.85 ± 0.47

Values are presented as mean ± standard deviation.

**Table 2 geriatrics-08-00041-t002:** Comparison of physical functions between subjects with the national average.

Physical Function	Sex	Age (Years)	*n*	Mean SD	National Average ^†^
Number of steps	Men	65–75	166	9027 ± 3422	6311
	Women	65–75	303	8484 ± 2873	5438
Grip strength (kg)	Men	65–69	71	34.6 ± 8.4	40.2
		70–75	93	33.4 ± 6.9	38.1
	Women	65–69	161	24.0 ± 5.5	25.3
		70–75	139	23.3 ± 5.5	23.9
OLS test (sec)	Men	65–69	70	78.6 ± 45.9	85.8
		70–75	94	62.7 ± 45.9	74.6
	Women	65–69	164	76.1 ± 43.2	89.0
		70–75	141	61.3 ± 42.6	74.3

^†^ National Health and Nutrition Survey 2017; OLS, one-leg standing test with eyes open.

**Table 3 geriatrics-08-00041-t003:** (**a**) Results from multiple linear regression models adjusted for sex examining the association of physical activities with physical functions ^1^. (**b**) Results from multiple linear regression models adjusted for sex and age examining the association of physical activities with physical functions ^2^. (**c**) Results from multiple linear regression models adjusted for sex, age and body mass index examining the association of physical activities with physical functions ^3^.

(a)																					
Physical Functions/Physical Activities		Number of Steps				Micro Exercise				Low Intensity Exercise				Medium Intensity Exercise				High Intensity Exercise			
	Variable	B	SE	β	*p*-value	B	SE	β	*p*-value	B	SE	β	*p*-Value	B	SE	β	*p*-Value	B	SE	β	*p*-Value
Muscle strength	Grip strength	7.29	22.7	0.019	0.748	0.000	0.000	−0.014	0.807	0.000	0.000	0.055	0.336	0.000	0.000	−0.043	0.456	0.000	0.000	−0.009	0.874
	Knee extension muscle strength	830.1	1099	0.036	0.450	0.012	0.015	0.035	0.447	−0.006	0.011	−0.025	0.588	−0.006	0.009	−0.032	0.492	0.000	0.001	0.004	0.941
Movement function	Two step test	4511	885	0.232	0.000 *	−0.038	0.013	−0.134	0.003 *	0.016	0.009	0.082	0.071	0.020	0.007	0.127	0.006 *	0.002	0.001	0.072	0.125
Agility function	Open-close stepping test	35.8	27.5	0.061	0.193	0.000	0.000	0.021	0.647	0.000	0.000	−0.019	0.679	0.000	0.000	−0.016	0.727	0.000	0.000	0.016	0.737
Balance function	OLS test (sedentary balance)	12.44	3.16	0.180	0.000 *	0.000	0.000	−0.093	0.038 *	0.000	0.000	0.080	0.077	0.000	0.000	0.045	0.328	0.000	0.000	0.145	0.002 *
	Functional reach test (movement balance)	18.9	21.1	0.042	0.371	0.000	0.000	−0.008	0.863	0.000	0.000	0.012	0.790	0.000	0.000	−0.014	0.757	0.000	0.000	0.079	0.090
Walking function	TUG test	−493	159	−0.143	0.002 *	0.002	0.002	0.033	0.469	−0.001	0.002	−0.035	0.434	0.000	0.001	−0.013	0.776	0.000	0.000	−0.007	0.879
**(b)**																					
**Physical Functions/Physical Activities**		**Number of Steps**				**Micro Exercise**				**Low Intensity Exercise**				**Medium Intensity Exercise**				**High Intensity Exercise**			
	**Variable**	**B**	**SE**	**β**	** *p* ** **-Value**	**B**	**SE**	**β**	** *p* ** **-Value**	**B**	**SE**	**β**	** *p* ** **-Value**	**B**	**SE**	**β**	** *p* ** **-Value**	**B**	**SE**	**β**	** *p* ** **-Value**
Muscle strength	Grip strength	−2.40	22.6	−0.006	0.915	0.000	0.000	0.011	0.840	0.000	0.000	0.029	0.607	0.000	0.000	−0.053	0.358	0.000	0.000	−0.026	0.662
	Knee extension muscle strength	438.9	1089	0.019	0.687	0.018	0.015	0.053	0.249	−0.010	0.011	−0.043	0.348	−0.007	0.009	−0.040	0.402	0.000	0.001	−0.007	0.881
Movement function	Two step test	4052	888	0.209	0.000 *	−0.030	0.013	−0.108	0.016 *	0.011	0.009	0.054	0.237	0.018	0.007	0.119	0.011 *	0.001	0.001	0.056	0.241
Agility function	Open-close stepping test	20.6	27.4	0.035	0.453	0.000	0.000	0.048	0.283	0.000	0.000	−0.048	0.294	0.000	0.000	−0.027	0.559	0.000	0.000	−0.001	0.985
Balance function	OLS test (sedentary balance)	10.49	3.20	0.152	0.001 *	0.000	0.000	−0.062	0.171	0.000	0.000	0.046	0.308	0.000	0.000	0.032	0.485	0.000	0.000	0.129	0.006 *
	Functional reach test (movement balance)	12.1	20.9	0.027	0.564	0.000	0.000	0.008	0.866	0.000	0.000	−0.004	0.933	0.000	0.000	−0.020	0.658	0.000	0.000	0.070	0.132
Walking function	TUG test	−413	159	−0.119	0.010 *	0.000	0.002	0.007	0.867	0.000	0.002	−0.010	0.819	0.000	0.001	−0.002	0.967	0.000	0.000	0.009	0.855
**(c)**																					
**Physical Functions/Physical Activities**		**Number of Steps**				**Micro Exercise**				**Low Intensity Exercise**				**Medium Intensity Exercise**				**High Intensity Exercise**			
	**Variable**	**B**	**SE**	**β**	** *p* ** **-Value**	**B**	**SE**	**β**	** *p* ** **-Value**	**B**	**SE**	**β**	** *p* ** **-Value**	**B**	**SE**	**β**	** *p* ** **-Value**	**B**	**SE**	**β**	** *p* ** **-Value**
Muscle strength	Grip strength	1.68	22.4	0.004	0.940	0.000	0.000	0.004	0.942	0.000	0.000	0.032	0.578	0.000	0.000	−0.044	0.444	0.000	0.000	−0.019	0.746
	Knee extension muscle strength	10.2	1091	0.000	0.993	0.022	0.015	0.067	0.146	−0.011	0.011	−0.049	0.300	−0.010	0.009	−0.057	0.230	−0.001	0.001	−0.020	0.687
Movement function	Two step test	3789	892	0.195	0.000 *	−0.028	0.013	−0.098	0.031 *	0.010	0.009	0.051	0.267	0.016	0.007	0.105	0.024 *	0.001	0.001	0.045	0.342
Agility function	Open-close stepping test	18.3	27.2	0.031	0.502	0.000	0.000	0.051	0.255	0.000	0.000	−0.048	0.286	0.000	0.000	−0.031	0.507	0.000	0.000	−0.003	0.941
Balance function	OLS test (sedentary balance)	9.42	3.21	0.136	0.003 *	0.000	0.000	−0.050	0.271	0.000	0.000	0.043	0.355	0.000	0.000	0.017	0.709	0.000	0.000	0.120	0.012 *
	Functional reach test (movement balance)	10.1	20.8	0.022	0.627	0.000	0.000	0.011	0.812	0.000	0.000	−0.005	0.917	0.000	0.000	−0.024	0.597	0.000	0.000	0.067	0.147
Walking function	TUG test	−380	158	−0.110	0.017 *	0.000	0.002	0.000	0.992	0.000	0.002	−0.008	0.860	0.000	0.001	0.007	0.880	0.000	0.000	0.015	0.751

^1^ Adjusted for sex; ^2^ Adjusted for sex and age; ^3^ Adjusted for sex, age and body mass index; * *p*-value < 0.05; B, partial regression coefficient; β, standardized partial regression coefficient; OLS, one-leg standing test with eyes open; SE, standard error; TUG, Timed up and go.

**Table 4 geriatrics-08-00041-t004:** (**a**) Results from multiple linear regression models adjusted for sex examining the association of physical functions with food intake ^1^. (**b**) Results from multiple linear regression models adjusted for sex and age examining the association of physical functions with food intake ^2^. (**c**) Results from multiple linear regression models adjusted for sex, age and body mass index examining the association of physical functions with food intake ^3^.

(a)													
Foods and Nutrients/Physical Functions		Two-Step Test				OLS Test				TUG Test			
	Variable	B	SE	β	*p*-Value	B	SE	β	*p*-Value	B	SE	β	*p*-Value
Foods	Vegetables	0.000	0.000	0.103	0.023 *	0.038	0.018	0.095	0.039 *	−0.001	0.000	−0.141	0.002 *
	Fruits	0.000	0.000	0.112	0.015 *	- ^#^	-	-	-	-	-	-	-
	Seeds	0.003	0.001	0.108	0.017 *	-	-	-	-	-	-	-	-
	Milk	-	-	-	-	-	-	-	-	−0.001	0.000	−0.100	0.034 *
	Beverages	0.000	0.000	0.116	0.010 *	-	-	-	-	-	-	-	-
	Cereals	-	-	-	-	−0.064	0.023	−0.131	0.006 *	0.001	0.000	0.105	0.026 *
	Egg	−0.001	0.000	−0.106	0.018 *	-	-	-	-	-	-	-	-
	Meat	-	-	-	-	−0.175	0.059	−0.137	0.003 *	-	-	-	-
Nutrients	K	-	-	-	-	-	-	-	-	0.000	0.000	−0.268	0.000 *
	Mg	-	-	-	-	0.136	0.034	0.194	0.000 *	-	-	-	-
	Vit B6	0.117	0.029	0.233	0.000 *	-	-	-	-	-	-	-	-
Nutritional composition	Fib/C	0.98	0.475	0.096	0.040 *	368	134	0.129	0.006 *	−7.54	2.65	−0.134	0.005 *
**(b)**													
**Foods and Nutrients/Physical Functions**		**Two-Step Test**				**OLS Test**				**TUG Test**			
	**Variable**	**B**	**SE**	**β**	** *p* ** **-Value**	**B**	**SE**	**β**	** *p* ** **-Value**	**B**	**SE**	**β**	** *p* ** **-Value**
Foods	Vegetables	0.000	0.000	0.115	0.010 *	0.045	0.018	0.111	0.015 *	−0.001	0.000	−0.155	0.001 *
	Fruits	0.000	0.000	0.128	0.005 *	- ^#^	-	-	-	-	-	-	-
	Seeds	0.003	0.001	0.102	0.023 *	-	-	-	-	-	-	-	-
	Milk	-	-	-	-	-	-	-	-	−0.001	0.000	−0.108	0.020 *
	Beverages	0.000	0.000	0.097	0.031 *	-	-	-	-	-	-	-	-
	Cereals	-	-	-	-	−0.054	0.022	−0.110	0.016 *	0.001	0.000	0.098	0.036 *
	Egg	−0.001	0.000	−0.111	0.013 *	-	-	-	-	-	-	-	-
	Meat	-	-	-	-	−0.158	0.058	−0.124	0.006 *	-	-	-	-
Nutrients	K	-	-	-	-	-	-	-	-	0.000	0.000	−0.281	0.000 *
	Mg	-	-	-	-	0.145	0.033	0.207	0.000 *	-	-	-	-
	Vit B6	0.109	0.023	0.216	0.000 *	-	-	-	-	-	-	-	-
Nutritional composition	Fib/C	1.15	0.470	0.112	0.015 *	430	132	0.150	0.001 *	−8.39	2.63	−0.149	0.001 *
**(c)**													
**Foods and Nutrients/Physical Functions**		**Two-Step Test**				**OLS Test**				**TUG Test**			
	**Variable**	**B**	**SE**	**β**	** *p* ** **-Value**	**B**	**SE**	**β**	** *p* ** **-Value**	**B**	**SE**	**β**	** *p* ** **-Value**
Foods	Vegetables	0.000	0.000	0.105	0.019 *	0.040	0.018	0.099	0.028 *	−0.001	0.000	−0.150	0.001 *
	Fruits	0.000	0.000	0.130	0.004 *	- ^#^	-	-	-	-	-	-	-
	Seeds	0.003	0.001	0.098	0.028 *	-	-	-	-	-	-	-	-
	Milk	-	-	-	-	-	-	-	-	−0.001	0.000	−0.101	0.033 *
	Beverages	0.000	0.000	0.092	0.040 *	-	-	-	-	-	-	-	-
	Cereals	-	-	-	-	−0.056	0.022	−0.114	0.012 *	0.001	0.000	0.101	0.031 *
	Egg	−0.001	0.000	−0.103	0.020 *	-	-	-	-	-	-	-	-
	Meat	-	-	-	-	−0.150	0.057	−0.117	0.009 *	-	-	-	-
Nutrients	K	-	-	-	-	-	-	-	-	0.000	0.000	−0.165	0.000 *
	Mg	-	-	-	-	0.113	0.032	0.161	0.000 *	-	-	-	-
	Vit B6	0.098	0.022	0.194	0.000 *	-	-	-	-	-	-	-	-
Nutritional composition	Fib/C	1.07	0.468	0.105	0.023 *	407	131	0.142	0.002 *	−8.17	2.63	−0.145	0.002 *

^1^ Extracted using a forward selection method by forced entry of sex; ^2^ Extracted using a forward selection method by forced entry of sex and age; ^3^ Extracted using a forward selection method by forced entry of sex, age and body mass index; ^#^ Not shown because they could not be detected by a forward selection method; * *p*-value < 0.05; B, partial regression coefficient; β, standardized partial regression coefficient; Fib/C, the ratio of fiber per carbohydrate; OLS, one-leg standing test with eyes open; SE, standard error; TUG, Timed up and go.

**Table 5 geriatrics-08-00041-t005:** (**a**) Results from multiple linear regression models adjusted for sex examining the association of serum Mg concentration with Mg intake, total fiber intake and Fib/C ratio ^1^. (**b**) Results from multiple linear regression models adjusted for sex and age examining the association of serum Mg concentration with Mg intake, total fiber intake and Fib/C ratio ^2^. (**c**) Results from multiple linear regression models adjusted for sex, age and body mass index examining the association of serum Mg concentration with Mg intake, total fiber intake and Fib/C ratio ^3^.

(a)				
	Serum Mg Concentration			
	B	SE	β	*p*-Value
Mg intake (mg/d)	- ^#^	-	-	-
Total fiber intake (g/d)	-	-	-	-
Fib/C	1.286	0.486	0.123	0.008 *
**(b)**				
	**Serum Mg Concentration**			
	**B**	**SE**	**β**	** *p* ** **-Value**
Mg intake (mg/d)	- ^#^	-	-	-
Total fiber intake (g/d)	-	-	-	-
Fib/C	1.365	0.487	0.130	0.005 *
**(c)**				
	**Serum Mg Concentration**			
	**B**	**SE**	**β**	** *p* ** **-Value**
Mg intake (mg/d)	- ^#^	-	-	-
Total fiber intake (g/d)	-	-	-	-
Fib/C	1.341	0.488	0.128	0.006 *

^1^ Adjusted for sex; ^2^ Adjusted for sex and age; ^3^ Adjusted for sex, age and body mass index; ^#^ Not shown because they could not be detected by a forward selection method; * *p*-value < 0.05; B, partial regression coefficient; β, standardized partial regression coefficient; Fib/C, the ratio of fibre per carbohydrate; Mg, magnesium; SE, standard error.

## Data Availability

The datasets generated or analysed during this study are available from the corresponding author upon reasonable request.

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
