# Peer review of "Relationships among Physical Activity, Physical Function, and Food Intake in Older Japanese Adults Living in Urban Areas: A Cross-Sectional Study"

_geriatrics, 2023, doi:10.3390/geriatrics8020041_

Round 1

Reviewer 1 Report

This is a cross-sectional study based on 469 older adults aged 65-75 years living in Tokyo included between 2017 and 2018.

Frame of the study: It is difficult to understand the context this paper is put into: “ To understand diet that supports the extension of healthy life expectancy, we studied the relationships among quality of life, physical activity, physical function and dietary intake. How this study supports the extension of healthy life expectancy is difficult to see.

The QOL part is confusing. The heading and aim talk only about physical activity, physical function and food intake, so why this QOL part? Can it be deleted? Overall, the paper is confusing with several investigated associations but without any adjustments for multiple testing og including QOL, which is not a part of the aim.

A flow chart is not necessary for excluding 4 persons.

An overall description of the study population is lacking. I need a section where you describe the overall investigations i.e., questionnaires, physical measures, blood tests and so on. Several measurements are performed, but it is difficult to get an overview of these. For instance, when did you do the test of physical functions? Who tested the participants? Did they get the questionnaires first?

Selection bias: You need to describe how you enrolled the participants into the study. You write about the generalizability of the study population in the limitation section – that the participants seem to be relatively active and living in urban areas. But how were they selected? Self-selection? And why did they have a high average systolic blood pressure?

The results should be stated objectively. So please delete the line page 5 lines 172-74 when you compare the values with another investigation of Japanese people – it should be moved to the discussion section.

Limitations: Your small sample size is a limitation that you do not mention. You did not find an association between physical function such as muscle strength and agility and physical activity. It may be due to the small sample size, but this is not mentioned anywhere. At least you should add that to your limitation sections.

Could you skip some of the tests and make a more focused paper? It seems that you have tested a lot of associations in a cross-sectional sample but without any specific aim. I think the paper could be approved by having a clear focus and limiting the tests to what is interesting in relation to your aim. I think that all the information about vitamins could be in a separate paper.

This is a cross-sectional design where you do not know the order of the exposure and the outcome, thus you cannot conclude that balancing food and nutrition may improve QOL through increased physical function and physical activity. You have not done a longitudinal study and you did not do mediation analyses. Your data do not support the conclusion.

Overall, I think the paper could be interesting by focusing the paper, reconsider the frame of the study with healthy life expectancy, cutting down on all the testing (and please remember to adjust for multiple testing), making a more interesting discussion where you discuss interesting/relevant findings in relation to other studies (and not just repeat the findings) and revise your conclusion so it is in agreement of what you can find taking the study design into account.

Author Response

To Reviewer 1

Thank you very much for your detailed and accurate comments. Please find our point-by-point responses below. We would be very grateful if you could review our revised manuscript.

Frame of study and the QOL part:

As you pointed out, our main objective was to examine the relationship between physical activity, physical function, and food intake. We also now mention the link between our study variables and life expectancy (lines 53–59). We apologize for the confusion caused by the inclusion of physical QOL, which we have now removed.

A flow chart:

We deleted the flow chart figure.

An overall description of the study:

We revised Fig. 1 to instead include the study schedule as opposed to the flow chart. In this section, we also describe the timing of measurement of each item (lines 100–112).

Selection bias:

We clarified our inclusion and exclusion criteria in our revised Participants section (lines 76–85).

The comment about high systolic blood pressure:

We added a comment about high systolic blood pressure to our revised Discussion section (lines 290–292).

Objective statement of results (page 5 lines 172–174 in the initial version)

As you pointed out, the relevant part was deleted because the Discussion section contains similar comments.

Limitations

We added a note to the limitations section that the sample size was small. We also mentioned that the relationships between muscle strength, agility, and physical activity still require elucidation (lines 364–367).

Simplifying and focusing a paper:

We revised for brevity where possible and corrected for redundancy. We also removed some citations. We revised our Discussion section considerably (lines 283–374).

Reviewer 2 Report

“Relationships among physical activity, physical function, and food intake in older Japanese adults living in urban areas: A cross-sectional study” by Fushimi and colleagues is a cross-sectional study to investigate the relationships between quality of life, physical activity, physical function, and food intake in 469 older adults living in metropolitan Tokyo.

Although the topic is of current interest to the scientific community, the manuscript needs minor revisions before publication in Geriatrics.

Introduction -  This section should be improved by better discussing the cited evidence and reporting objective data to enable the reader to understand the current situation regarding this hot topic. Furthermore, the introduction is largely focused on diet, while the other parameters considered, such as physical activity and physical function, are poorly argued. For example, what type of physical activity is referred to (endurance, strength, mechanical vibration)? How is muscle function assessed and what evidence is there for it? In the literature, there is plenty of evidence on this (e.g. doi: 10.3390/jcm10122597; doi: 10.3892/br.2020.1283.; doi: 10.1186/s12905-019-0864-5; doi: 10.1016/S2666-7568(21)00079-9).

Otherwise, I have no further comments to add. The manuscript is well-written and well-structured, and I believe that improving the introduction is a key step in accepting the manuscript. 

Author Response

To Reviewer 2

Thank you very much for your careful peer review. Based on your advice, we added sentences to the Introduction section and cited some relevant studies. We would be very grateful if you could review our revised manuscript.

Introduction:

We added sentences mentioning physical activity, physical function, and their relationships to our revised Introduction section (lines 53–59).

Round 2

Reviewer 1 Report

The authors have addressed most of my comments and concerns from the first review; however, not the multiple testing part. You perform several tests in a cross-sectional sample, so considering multiple testing should be done. Please add a star to the p-values in Tables 3-5, where the p-values are significant after adjusting for multiple testing and add a sentence about the multiple testing part and the results in the manuscript. 

Author Response

To Reviewer 1

Thank you very much for your detailed comments.

Based on your comments, we have added tables for each adjustment items (Table 3-1, 3-2, 3-3, 4-1, 4-2, 4-3, 5-1, 5-2 and 5-3). The results were similar regardless of the adjustment items, and we have made the necessary comments in the “2.9 Statistical analysis”, “3 Results” and ”4 Discussion” section (lines 234-236, 261-262, 284-285 and 344-347). We apologize for displaying different results for the analysis method in Table 5. We would like to inform you that we have replaced them with the relevant tables.

We would be very grateful if you could review our re-revised manuscript.

Round 3

Reviewer 1 Report

I think the paper has improved during the review process and can now be accepted for publication; though I suggest that the paper go through a language/spell check before publication.